# Fungal Pathogens as Causes of Acute Respiratory Illness in Hospitalized Veterans: Frequency of Fungal Positive Test Results Using Rapid Immunodiagnostic Assays

**DOI:** 10.3390/jof9040456

**Published:** 2023-04-08

**Authors:** Diego H. Caceres, Maria C. Rodriguez-Barradas, Michael Whitaker, Brendan R. Jackson, Lindsay Kim, Diya Surie, Bryanna Cikesh, Mark D. Lindsley, Orion Z. McCotter, Elizabeth L. Berkow, Mitsuru Toda

**Affiliations:** 1Centers for Disease Control and Prevention (CDC), Atlanta, GA 30329, USA; 2Center of Expertise in Mycology Radboudumc, Canisius Wilhelmina Hospital, 6532 SZ Nijmegen, The Netherlands; 3Studies in Translational Microbiology and Emerging Diseases (MICROS) Research Group, School of Medicine and Health Sciences, Universidad del Rosario, Bogota 111221, Colombia; 4Michael E. DeBakey VA Medical Center, Houston, TX 77030, USA; 5US Public Health Service, Rockville, MD 20852, USA; 6Oregon Health Authority, Portland, OR 97232, USA

**Keywords:** *Coccidioides*, *Histoplasma*, *Aspergillus*, acute respiratory illness, laboratory testing

## Abstract

Fungal respiratory illnesses caused by endemic mycoses can be nonspecific and are often mistaken for viral or bacterial infections. We performed fungal testing on serum specimens from patients hospitalized with acute respiratory illness (ARI) to assess the possible role of endemic fungi as etiologic agents. Patients hospitalized with ARI at a Veterans Affairs hospital in Houston, Texas, during November 2016–August 2017 were enrolled. Epidemiologic and clinical data, nasopharyngeal and oropharyngeal samples for viral testing (PCR), and serum specimens were collected at admission. We retrospectively tested remnant sera from a subset of patients with negative initial viral testing using immunoassays for the detection of *Coccidioides* and *Histoplasma* antibodies (Ab) and *Cryptococcus, Aspergillus*, and *Histoplasma* antigens (Ag). Of 224 patient serum specimens tested, 49 (22%) had positive results for fungal pathogens, including 30 (13%) by *Coccidioides* immunodiagnostic assays, 19 (8%) by *Histoplasma* immunodiagnostic assays, 2 (1%) by *Aspergillus* Ag, and none by *Cryptococcus* Ag testing. A high proportion of veterans hospitalized with ARI had positive serological results for fungal pathogens, primarily endemic mycoses, which cause fungal pneumonia. The high proportion of *Coccidioides* positivity is unexpected as this fungus is not thought to be common in southeastern Texas or metropolitan Houston, though is known to be endemic in southwestern Texas. Although serological testing suffers from low specificity, these results suggest that these fungi may be more common causes of ARI in southeast Texas than commonly appreciated and more increased clinical evaluation may be warranted.

## 1. Introduction

Pneumonia was the ninth leading cause of death in the United States before the COVID-19 pandemic, resulting in ~50,000 deaths annually and responsible for >1.5 million emergency department visits in 2018 [1,2]. Viral infections are thought to be the most common cause of pneumonia, led by influenza, respiratory syncytial virus (RSV), and rhinovirus infections [3]. Bacteria are also common causes of pneumonia, particularly *Streptococcus pneumoniae* and *Mycoplasma pneumoniae*, accounting for ~14% of community-acquired pneumonia (CAP) requiring hospitalization in adults in the United States [4]. Acute respiratory infections or illnesses (ARI) constitute a broader range of conditions beyond CAP, which are leading causes of childhood mortality globally [5,6].

Fungal infections are established causes of ARI and CAP, but their frequency is generally not well defined, and it is rarely possible to clinically distinguish these infections from viral and bacterial infections [7,8,9,10,11]. Nevertheless, fungal infections are infrequently considered or tested for by clinicians who care for ARI or CAP patients. For example, the largest U.S. CAP study to date, involving 2259 patients, tested for a wide range of viral and bacterial pathogens but did not include systematic fungal testing; despite the lack of inclusion, clinician-ordered tests detected *Coccidioides* and *Histoplasma* infections [4]. In a 2020 survey among U.S. healthcare providers, <4% reported frequently testing CAP patients for coccidioidomycosis or histoplasmosis, with some variability by geography [9]. Although fungal diagnosis guidelines exist, the paucity of testing is consistent with the Infectious Disease Society of America and American Thoracic Society guidelines for CAP, which do not include fungal diseases [12,13]. Since the burden of fungal ARIs remains largely unknown, the relevance of testing in most regions is also unclear. Among the few places in which the contribution of fungal infections to CAP has been assessed are southern Arizona and southern California, where coccidioidomycosis has been found to be a leading cause of pneumonia, accounting for 15–30% of CAP cases and involving thousands of cases annually [14,15,16,17]. 

However, even in endemic areas where CAP patients often initially present, providers infrequently test for coccidioidomycosis, resulting in delayed diagnosis and appropriate treatment [9,11,18,19]. Testing is likely even more uncommon across the disease’s far wider endemic area, which likely encompasses much of the western United States, including much of Texas, as well as northern Mexico and other parts of Latin America, where serological diagnostics may not be available. Histoplasmosis occurs across much of the central and eastern United States, including eastern Texas, and increasing evidence suggests it also occurs across the country and the world, frequently undetected and misdiagnosed [20,21]. When cases are detected, it is often incidentally, when *Histoplasma* is identified by culture or on biopsy, rather than through directed antigen or antibody testing [22]. Diagnosis with coccidioidomycosis and histoplasmosis typically follows multiple healthcare provider visits for symptoms and 2–3 courses of ineffective antibiotics, which pose risks of adverse effects, microbiome derangement, and resistance development [9,11,16,18,19]. 

Other fungi can also cause lung infections, including *Cryptococcus* species, primarily the *C. neoformans* and *C. gattii* species complexes, and mold such as *Aspergillus* species [23,24,25,26,27]. Their contribution to the overall ARI burden is unclear, although thought to be low. However, *Aspergillus* species are a frequent cause of lung infections in severely immunocompromised patients and have been increasingly documented as co-infections in severely ill patients with or without influenza and COVID-19 [28,29,30,31,32]. 

We aimed to examine the prevalence of positive fungal immunodiagnostics for often-overlooked fungal diseases as potential causes of ARI in hospitalized patients with negative viral or bacterial testing. We analyzed remnant serum specimens collected as part of a larger ARI surveillance at a large reference Veterans Affairs Medical Center (VAMC) with a patient catchment from Houston, Texas, and surrounding areas.

## 2. Materials and Methods

*Patient population:* The Surveillance Platform for Enteric and Respiratory Infectious Organisms at the VA (SUPERNOVA) is a network of five U.S. Veterans Affairs Medical Centers (VAMC) that conducts active and passive surveillance for ARI, and includes laboratory testing for viral pathogens [33]. Our fungal serologic study examined remnant sera that were collected as part of the larger SUPERNOVA study during November 2016–August 2017 at the Michael E. DeBakey VAMC (MEDVAMC) in Houston, Texas. For this study, the remnant sera were collected from subset of patients enrolled in the SUPERNOVA study.

ARI was defined broadly, and patients were eligible for inclusion if they were admitted for <72 h with any of the following symptoms or syndromes:
Influenza-like disease, influenza, upper respiratory infection (URI), viral URI, cough, or bronchitis;Pneumonia, bacterial pneumonia, community-acquired pneumonia, aspiration pneumonia, rule-out pneumonia, evaluate pneumonia, or bibasilar pneumonia;Chronic obstructive pulmonary disease (COPD) exacerbation, asthma exacerbation, status asthmaticus, or asthmatic bronchitis;Acute respiratory distress syndrome (ARDS), fever, nasal congestion, chest congestion, sore throat, chills, body aches/myalgias, fatigue, respiratory distress, shortness of breath, difficulty in breathing, dyspnea, sepsis, cystic fibrosis exacerbation, respiratory medical other, congestive heart failure, idiopathic pulmonary fibrosis, altered mental status and new onset, exacerbation, or change of two or more of the following symptoms with at least one respiratory symptom beginning less than 10 days;
oRespiratory symptoms: cough, shortness of breath, nasal congestion, chest congestion, or sore throat;oConstitutional symptoms: fever/feverishness, chills, body aches/myalgias, or fatigue.


Patients were excluded if they were >72 h from admission, transferred from another hospital after an admission of >48 h, had ARI duration of >10 days, or were previously enrolled in SUPERNOVA within the previous 30 days.

Following enrollment, SUPERNOVA study staff collected specimens from enrolled patients within 72 h of admission. Specimens included a mid-turbinate nasal swab, oropharyngeal (OP) swab, and a serum specimen obtained from venous blood phlebotomy. Mid-turbinate nasal and OP swabs were obtained using flocked swabs and placed in universal transport media (UTM) for molecular detection of pathogens. One swab was placed into a single nostril to collect epithelial cells and absorb secretions, and a second swab was used to swab the posterior pharynx. For intubated patients, a tracheal aspirate sample or bronchoalveolar lavage (BAL) was considered an acceptable alternative to mid-turbinate and OP swabs. The swabs placed in UTM were stored at 2–8 °C until processing (within 72 h).

Data collection: Basic demographic, laboratory, and clinical information were collected for each enrolled patient. Surveillance personnel used a standardized case report form (CRF) to abstract data from medical records. Data were abstracted into a REDCap database with restricted access to individuals with secure login credentials [34]. No patient identifiers were linked to CRF data or specimens. 

Laboratory testing: The MEDVAMC Supernova research team tested respiratory specimens at the time of enrollment using BioFire (BioFire Diagnostics, LLC, Salt Lake City, UT, USA), which tested for viruses such as influenza, RSV, parainfluenza, human metapneumovirus, rhinoviruses/enteroviruses, and adenovirus, and bacteria such as *Bordetella parapertussis*, *Bordetella pertussis*, *Chlamydia pneumoniae*, and *Mycoplasma pneumoniae*. Specimens were also tested for bacterial culture. The CDC Mycotic Diseases Branch (MDB) Laboratory retrospectively performed fungal immunodiagnostic assays on remnant serum specimens on a subset of patients. 

Coccidioides antibody (Ab) testing: We used the clarus *Coccidioides* Ab enzyme immunoassay (EIA) (CAb EIA) (Product reference CAB102, IMMY^®^, Norman, OK, USA) for the detection of IgM and IgG Ab, and the sōna *Coccidioides* Ab lateral flow assay (LFA) for the detection of total Ab (CAb LFA) (Product reference CTA2003, IMMY^®^, Norman, OK, USA). EIA results were grouped into three categories: positive CF (IgG) or TP (IgM) ≥ 1.5; indeterminate CF or TP ≥ 1–< 1.5; negative CF or TP < 1. Specimens were also tested by immunodiffusion (ID) for the detection of anti-*Coccidioides* IgG (CAb ID) in a CLIA-certified laboratory at MDB.

Histoplasma antigen (Ag) and Ab testing: We used the clarus *Histoplasma* galactomannan EIA (HisAg EIA) (Product reference HGM201, IMMY^®^, Norman, OK, USA), which detects *Histoplama* Ag in urine specimens. Since urine specimens were unavailable for this study, we validated the antigen test in serum specimens. The validation was completed using a standardized reference panel of serum specimens collected as part of a prospective study [35] (Appendix A). We used the same cutoff for urine specimens (0.2 ng/mL) based on this validation. Specimens were stored at −80 °C and were prepared and pretreated according to the protocol for the Bio-Rad^®^ Platelia *Aspergillus* kit; this pretreatment reduces the likelihood of false positive results. The supernatant of the pretreated serum was tested according to the HisAg EIA manufacturer’s instructions. Serum specimens were also tested using MDB’s anti-*Histoplasma* Ab ID assay (HisAb ID). For HisAb ID, we looked for the presence of two precipitins against the M and H antigens (H and M bands).

Cryptococcus and Aspergillus Ag testing: *Cryptococcus* Ag testing was performed using a commercial LFA kit (CrAg^®^ LFA. Product reference CR2003, IMMY^®^, Norman, OK, USA) and *Aspergillus* Ag using the Platelia *Aspergillus* kit (AspAg EIA) (Product reference 62794, Bio-Rad^®^, Hercules, CA, USA). Testing was performed according to the manufacturer’s instructions.

Case definition: We defined patients who tested positive for any of the fungal serologic tests as those with positive fungal testing (i.e., Cab EIA, Cab LFA, Cab ID, HisAg EIA, HisAb ID, CrAg LFA, and AspAg EIA). *Coccidioides* antibody tests are commonly used for the diagnosis of recent and/or active infection, although cross-reactivity with *Histoplasma* and *Blastomyces* cannot be ruled out. *Histoplasma* antibody tests could help increase the specificity of *Histoplasma* infection, although prior infections and cross-reactivity with *Blastomyces* cannot be ruled out. We grouped those with no positive fungal test results as patients with negative fungal test results. 

Statistical analysis: Characteristics of patients with positive or negative results by fungal testing were examined. Odds ratios (OR) and 95% confidence intervals (CI) were calculated using binomial logistic regression with *p* < 0.05 considered statistically significant. Analyses were performed using the software EPIDAT 3.1 and Stata 11.0.

Patient consent statement: This study was reviewed and approved by the MEDVAMC Research and Development Committee (VA ID: 15J03.HB), Baylor College of Medicine (IRB #: H-37327) and the Centers for Disease Control and Prevention. Written consent was obtained from study participants. All clinical information from the participants in the study were anonymized in a database using an alphanumerical code.

## 3. Results

During November 2016–August 2017, 328 hospitalized patients met the criteria for ARI on admission. We received remnant sera for 224 (68%) ARI hospitalized patients. 

Of the 224 ARI patients in our investigation cohort, 22% (49/224) tested positive by one or more of the fungal immunodiagnostic assays. Thirty (13%) patient specimens tested positive by *Coccidioides* antibody tests, nineteen (8%) by *Histoplasma* immunodiagnostic assays, two (1%) by *Aspergillus* Ag, and none by *Cryptococcus* Ag testing; two (1%) tested positive by both *Coccidioides* and *Histoplasma* immunodiagnostic assays (Figure 1). Ten patients with positive fungal results also had results indicative of primary viral (seven influenza) or bacterial infections (two bacteremia). Two additional patients grew coagulase-negative *Staphylococcus* in blood cultures that were considered contaminants (Table 1).

Almost all the patients were men (96%) with a median age of 68 years old (interquartile range 62–73 years). Three quarters (77%; n = 172) of patients received antibacterial medications, and 8% (n = 18) received antiviral treatments; no patient received antifungal medications. Fifteen percent of patients were hospitalized in the intensive care unit (ICU), and 5% were intubated for mechanical respiratory support; 218 (97%) were discharged alive. No significant differences were observed in the demographic characteristics between patients who had a positive fungal test compared to those with a negative fungal test (Table 1). Diarrhea was associated with patients who had positive fungal results (OR 2.03, 95% CI 1.04–3.95). Fatigue (OR 0.42, CI 0.19–0.93), headache (OR 0.49, CI 0.25–0.98), and productive cough (OR 0.39, CI 0.19–0.81) were associated with patients with negative fungal results (Table 1). 

*Coccidioides:* Of the thirty patients that tested positive by *Coccidioides* Ab, thirteen tested positive by Cab LFA; ten of these thirteen patients had indeterminate results by Cab EIA. Eight were positive by Cab EIA only (one positive by IgM EIA and seven positive by IgG EIA). Six were positive by both Cab LFA and Cab EIA (all IgG) and one positive by CLIA ID (Figure 1). The EIA-IgM-positive patient had an EIA index of 3.5 and IgM-positive patients had EIA index ranging 1.5–3.9. Patients with positive *Coccidioides* Ab presented with similar demographic and clinical features compared to those with a negative fungal test result. Three patients were admitted to the ICU; all patients were discharged alive (Table 1).

*Histoplasma:* Nineteen (8%) patients had a positive *Histoplasma* immunodiagnostic result, including sixteen (7%) by HisAg EIA for Ag with *Histoplasma* Ag concentrations ranging 0.2 ng/mL–5.4 ng/mL, and three (1%) by *Histoplasma* ID for Ab. Three specimens yielded an M band and one yielded both an H and M band. Two specimens (one by EIA and one by ID) also had positive tests for *Coccidioides*. We found no significant differences in demographic characteristics between patients with positive *Histoplasma* results and patients with a negative fungal test (Table 1). Two of the six patients also tested positive for *Coccidioides*. Four patients were admitted to the ICU; all patients survived to discharge (Table 1). 

*Aspergillus:* Two patients (1%) tested positive by the AspAg EIA, one with an index of 1.3, and another with an index of 0.5 (Figure 1). One patient received ICU care (Table 1). Both patients were discharged alive.

## 4. Discussion

In this investigation of veterans hospitalized with ARI from southeastern Texas, a high proportion of hospitalized ARI patients had positive serological results for fungal pathogens that can cause pneumonia, nearly all (96%, 47/49) of which were for *Coccidioides* or *Histoplasma*. Fourteen percent of patients with a positive fungal test had severe disease requiring ICU admission. Although serological testing suffers from the possibility of cross-reactivity and potential false positivity, the results suggest that these fungi may be more common causes of ARI in southeast Texas than commonly appreciated. Since none of these patients were specifically interviewed for endemic fungi risk factors, tested for, or diagnosed with these infections while hospitalized, we cannot assess for past infection and/or if the clinical presentation was related to a fungal infection. Further assessment of endemic mycoses as causes of ARI and CAP in hospitalized patients is warranted, since a lack of fungal disease testing can lead to under-recognition, misdiagnosis, inappropriate antibiotic use, lack of antifungal treatment, and poorer outcomes [7,9,10,15,16,36].

Distinguishing symptoms of fungal versus non-fungal infections is challenging. In our comparisons of individuals with positive and negative fungal results, we did not find significant differences between most of the signs and symptoms that ARI patients displayed. This finding is similar to the results observed in a study that compared symptoms of patients with endemic mycoses versus other respiratory illnesses in commercially insured adult outpatients in the United States and other studies [7,14,37,38].

Using both Ag and Ab tests increased the detection of possible histoplasmosis, consistent with other studies [35,39,40,41,42]. It is possible that some of these infections may have resulted in non-clinically significant presentations or may reflect cross-reactivity with *Coccidioides*. However, given that *Histoplasma* testing is highly specific, with fifteen patients hospitalized and four admitted to the ICU, our analysis suggests that the Houston area may be more affected by histoplasmosis than conventionally considered [43,44,44,45,46,47].

A high proportion (13%) of ARI cases with positive *Coccidioides* Ab results were seen in this cohort in the Houston area that is not traditionally considered endemic for coccidioidomycosis. This proportion is only slightly lower than the ~15–30% of CAP seen in highly endemic areas. It is possible that some of these cases reflect false positives given the limitations of *Coccidioides* serology, including cross-reactivity with *Histoplasma* or *Blastomyces* [48,49,50,51], supported by a low number with positive immunodiffusion results. However, the immunodiffusion antibody assay has low sensitivity, particularly early in infection. That half of the positive *Coccidioides* results were by LFA alone also raises questions about false positivity, as LFA performance data are still limited [52]. Still, nearly half were positive by EIA, which is used widely for diagnosis in endemic areas such as Arizona and typically has good specificity (>85%), although the positive predictive value (46–90%) can vary depending on the test specificity [48,49,50].

It is possible that some positive *Coccidioides* tests, particularly for IgG alone, represent past infection; however, *Coccidioides* IgG does not persist long after the initial infection and is often used for the diagnosis of recent infections [53,54]. Houston is >100 miles east of areas of Texas with documented locally acquired coccidioidomycosis cases of an outbreak in Beeville, Texas [55]; it is also much farther northeast and much wetter than areas considered highly endemic in West Texas and along the Rio Grande Valley. However, the endemic area for coccidioidomycosis remains poorly defined and may be changing over time. Lack of travel history of these cases also limits interpretations. Further, coccidioidomycosis is not reportable in Texas, limiting our understanding of its geography and frequency of occurrence. Further assessment of coccidioidomycosis as a potential cause of ARI in Houston is warranted.

This report has several limitations, particularly the potential for false positives and cross-reactivity, as well as the inability to determine whether fungal infections, when present, were the cause of the patient’s symptoms. In addition, the study only included a single medical institution in Houston, Texas, which limits the generalizability of the results. We also could not evaluate immune conversion [51] because we did not have access to paired specimens, resulting in lower specificity.

Despite the limitations, the findings from our investigation suggest that fungal infections may be more common than appreciated in ARI, particularly since viral or bacterial etiologies were not identified in most cases. It highlights the importance for clinicians to consider and test for fungal diseases in patients with ARI, especially for those who test negative for viral or bacterial infections, to increase timely treatment and to reduce inappropriate use of antibiotics. The best diagnostic testing modality in this setting needs to be further evaluated.

## Figures and Tables

**Figure 1 jof-09-00456-f001:**
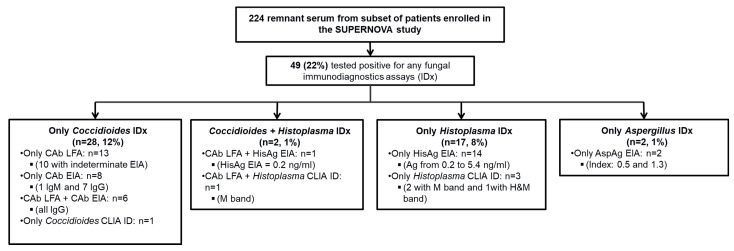
Flow chart of specimens tested and results. (ID) immunodiffusion; (CAb LFA) Sōna *Coccidioides* Ab lateral flow assay; (CAb EIA) Clarus *Coccidioides* Ab enzyme immunoassay (CAb EIA); (HisAg EIA) Clarus *Histoplasma* galactomannan EIA (HisAg EIA); (AspAg EIA) Platelia *Aspergillus* kit.

**Table 1 jof-09-00456-t001:** Characteristics of 224 hospitalized patients with acute respiratory illness.

Characteristics	All n (%)	Positive Fungal Testing n (%)	Negative Fungal Testing n (%)	OR (CI)	*p*
Total	224	49	175		
Median age (IQR)	68 (11)	68 (9)	68 (12)	-	-
Sex (male)	214 (96)	47 (96)	167 (95)	0.88 (0.18–4.23)	0.883
Race white	128 (57)	26 (53)	102 (58)	0.80 (0.42–1.52)	0.514
Race black	92 (41)	23 (47)	69 (39)	1.35 (0.71–2.57)	0.346
Race unknown	4 (2)	0 (0)	4 (3)	1.23 (0.65–2.33)	0.514
Ethnicity hispanic	13 (6)	4 (8)	9 (5)	1.63 (0.48–5.56)	0.428
**Constitutional symptoms**					
Fatigue	**191 (85)**	**37 (76)**	**154 (88)**	**0.42 (0.19–0.93)**	**0.033 ***
Loss of appetite	127 (57)	25 (51)	102 (59)	0.73 (0.38–1.38)	0.343
Chills	112 (50)	22 (45)	90 (51)	0.76 (0.41–1.45)	0.420
Myalgias	97 (43)	22 (45)	75 (43)	1.08 (0.57–2.05)	0.799
Headache	**92 (41)**	**14 (29)**	**78 (45)**	**0.49 (0.25–0.98)**	**0.047 ***
Fever	83 (41)	16 (38)	67 (42)	0.85 (0.42–1.71)	0.658
Confusion	69 (31)	12 (24)	57 (33)	0.66 (0.32–1.37)	0.271
Earache	20 (9)	2 (4)	18 (10)	0.36 (0.08–1.64)	0.191
Conjunctivitis	17 (8)	4 (8)	13 (7)	1.10 (0.34–3.56)	0.864
Skin rash	11 (5)	4 (8)	7 (4)	2.12 (0.59–7.56)	0.247
**Respiratory symptoms**					
Cough	212 (95)	48 (98)	164 (94)	2.92 (0.36–23.44)	0.312
Productive cough	**137 (83)**	**32 (67)**	**137 (84)**	**0.39 (0.19–0.81)**	**0.012 ***
Dyspnea	208 (93)	45 (92)	163 (93)	0.83 (0.25–2.69)	0.754
Wheezing	160 (71)	31 (63)	129 (74)	0.61 (0.31–1.20)	0.155
Nasal congestion	123 (59)	27 (55)	105 (60)	0.80 (0.42–1.52)	0.510
Respiratory rate	107 (48)	27 (55)	80 (45)	1.44 (0.76–2.72)	0.260
Chest retractions	95 (42)	18 (37)	77 (44)	0.73 (0.38–1.41)	0.364
Chest pain	89 (40)	16 (33)	73 (42)	0.66 (0.34–1.29)	0.231
Sore of throat	77 (35)	17 (35)	60 (34)	1.00 (0.51–1.96)	0.978
**Gastrointestinal symptoms**					
Diarrhea	**64 (29)**	**20 (41)**	**44 (25)**	**2.03 (1.04–3.95)**	**0.036 ***
Abdominal pain	51 (23)	14 (29)	37 (21)	1.48 (0.72–3.03)	0.284
Vomiting	23 (10)	8 (16)	15 (9)	2.06 (0.82–5.21)	0.123
**Underlying conditions**					
Heart disease	176 (79)	38 (80)	138 (79)	1.04 (0.47–2.28)	0.911
Genetic disorder	118 (53)	26 (53)	92 (53)	1.01 (0.54–1.92)	0.952
Neurologic disease	110 (49)	25 (51)	85 (49)	1.10 (0.58–2.07)	0.762
Diabetes	102 (46)	21 (43)	81 (46)	0.87 (0.45–1.64)	0.670
Emphysema	98 (44)	16 (33)	82 (47)	0.54 (0.28–1.07)	0.079
Oxygen support	84 (38)	20 (41)	64 (37)	1.18 (0.62–2.26)	0.607
Tobacco smoker	71 (32)	21 (43)	50 (29)	1.87 (0.97–3.60)	0.060
Cancer	61 (27)	16 (33)	45 (26)	1.40 (0.70–2.78)	0.336
Kidney disease	51 (23)	15 (31)	36 (31)	1.70 (0.84–3.46)	0.141
Immunodeficiency	38 (17)	9 (18)	29 (17)	1.13 (0.49–2.58)	0.767
Liver disease	25 (11)	6 (12)	19 (11)	1.14 (0.43–3.04)	0.785
Asthma	17 (8)	4 (8)	13 (7)	1.10 (0.34–3.56)	0.864
Transplant	5 (2)	3 (6)	2 (1)	5.64 (0.91–34.76)	0.062
**Treatments**					
Antibiotics	172 (77)	38 (78)	134 (77)	1.05 (0.49–2.25)	0.886
Antivirals	18 (8)	5 (10)	13 (7)	1.41 (0.48–4.18)	0.529
**Hospital encounters and outcomes**					
ICU admission	34 (15)	7 (14)	27 (15)	0.91 (0.37–2.24)	0.844
Intubated	11 (5)	2 (4)	9 (5)	0.78 (0.16–3.75)	0.762
Outcome (survived)	218 (97)	49 (100)	169 (97)	-	-

Negative viral testing: respiratory specimens that tested negative using approved BioFire (BioFire Diagnostics, LLC, Salt Lake City, UT, USA) for respiratory pathogens, including influenza, RSV, parainfluenza, human metapneumovirus, rhinoviruses/enteroviruses, and adenovirus. (*) *p* < 0.05, OR: odds ratio, CI: 95% confidence interval.

## Data Availability

The data that support the findings of this study are available from the corresponding author upon reasonable request.

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
