# Peer review of "Fungal Pathogens as Causes of Acute Respiratory Illness in Hospitalized Veterans: Frequency of Fungal Positive Test Results Using Rapid Immunodiagnostic Assays"

_jof, 2023, doi:10.3390/jof9040456_

Round 1
Reviewer 1 Report
The study is original in the sense that there are no other published papers investigating the frequency of fungal positive immunodiagnostic tests in a specific US area, southeastern Texas. Some findings of the study are important because they indicate a potential higher incidence of fungal respiratory infections than expected in this area.
Overall, the resources and data used were rich but the manuscript lacks of clarity so it is possible that all this information is not well presented and discussed.
It is noteworthy that the authors acknowledge the limitations of the study, some of which are inherent to the diagnostic tests and not only to the study design.
I suggest that the manuscript be reviewed and ordered so that the methodology and the results are clear and compelling.
I present the following comments in order to help improve the manuscript.
Major comments:
Materials and Methods: It is not clear if the study was retrospective, using remnant serum from a previous study or if it was a transversal study performed in parallel or included in a larger SUPERNOVA study. From paragraph between L134 and L136 the first situation seems more likely. Please clarify this point.
Laboratory testing: All patients were tested with BioFire and culture? Only blood cultures (table 1)? Second viral testing (table 1)?
The aim of the study states that the population study were patients with negative viral or bacterial testing. It would be interesting to know if there were fungal and viral and/or bacterial co-infections. The strategy you propose excludes this possibility, although finally some (or all?) patients underwent a second round of testing, not mentioned in materials and methods but described in results. All this points should be clarified. On the other hand, in L96-97 you mention that SUPERNOVA conducts surveillance of ARI including laboratory testing for viral pathogens; no bacterial pathogens mentioned here, (although BioFire tests for some of them. Nevertheless, in Laboratory testing section you say that “specimens were also tested for bacterial culture”. Which specimens were those? And what kind of cultures were performed?
L181-188: Confusing sentences for case definitions. Explain what is the relevance of mentioning possible cross-reactivity between tests. Did it have a consequence in patients' classification? If not, this should be mentioned along with the description of the tests or in limitations of the study. All the "Case definition" item should be reformulated.
L209-211: In objectives and methods it is stated that fungal testing will not be performed to patients with positive viral or bacterial results.
Minor comments:
L42-44: Other bacteria can also cause CAP and they were found in the cited reference (Ref 4). I would be also worth mentioning that this study investigated patients with CAP who required hospitalization, same as your study.
L57-58: It should be noticed that the American Thoracic Society published a clinical practice guideline for microbiological laboratory testing in the diagnosis of fungal infections in pulmonary and critical care practice.
Results
Coccidioides (L233-243): It seems that patients who tested “indeterminate” on EIA were classified as positive. Not discussed in the "Discussion" section.
Histoplasma: it not clear, particularly in reference to potential co-infections.
L250-251 – How are these results interpreted? Not specified in materials and methods nor in results.
L256-257 – “Two of the six patients also tested positive for Coccidioides”. Are the six patients those who tested positive for Influenza? In that case, one of them did not test positive for Histoplasma, according to table 1.
L271-272: “One in eight patients…” Where does this figure come from?
Table 1: Title is not clear enough. Something like “Description of patients with concomitantly positive results in fungal and viral or bacterial tests” would be more appropriate to be strict. Or simply “Description of patients with potential co-infections detected by laboratory testing” assuming that patients with positive results could be actually infected.
Table 2: Title is not accurate as the table includes 10 patients who did test positive for viral or bacterial agents.
Reviewer 2 Report
Dear authors:
Here I am sending a few doubts and clarifications about the article:
Line 103: It says: Influenza-like illness, influenza-like disease. That is the same.
The authors performed the detection of Coccidioides antibodies by EIA, LFA and also carried out immunodiffusion test, but did not perform the antigen detection of this pathogen. On the other hand, they tested sera for the detection of both antibodies and Histoplasma antigens. Couldn’t it be useful to also test Coccidioides antigens in this context?
The authors say that they only studied sera from patients that did not test positive for viral or bacterial pathogens. But on lines 209-211 they found that 10 patients with positive results for fungal infections also had results indicative of primary viral or bacterial infections. Could you explain this?
Figure 1 shows that there were 10 patients with indeterminate EIA for Coccidioides. How did you interpret these results?
Table 1 describes ten cases with fungal diagnosis and positive results for bacterial or viral infections. Did these viral or bacterial diagnoses precede or follow the fungal diagnosis?
Line 261. It saysthat 2 patients had positive AspAg EIA results with index of 1.3 and 0.5 but Figure 1 says 1.3 and 0.3.
Line 261. It says that 2 patients had positive AspAg EIA results with index of 1.3 and 0.5 but Figure 1 says 1.3 and 0.3.
Round 2
Reviewer 1 Report
The paper has been much improved and the methodology is clear now. The authors make a huge effort to deepen the knowledge of the epidemiology of fungal respiratory infections, a subject not much investigated, and the conclusions are relevant.
Nevertheless, the paper could be further improved by
L24-25: Abstract: patients who initially tested negative for viral or bacterial causes? Contradictory to what is described in Laboratory methods. Actually, it was deleted from the original draft (L155). This inconsistency needs to be corrected.
L102: Please clarify if the subset of patients consisted in those from whom remnant serum could be obtained (deduced from what is stated in L208-209) or if other criteria were applied.
L312: Still not explained how to interpret the 10 indeterminate LFA results, which account for a third of patients who tested positive for Coccidioides.
